# A joint research agenda for climate action bridges behavioral sciences and urban planning

Felix Creutzig & Lucia A. Reisch

We argue for systematically integrating behavioral sciences and urban planning to develop a joint agenda for research and planning practice. By viewing urban form as a critical choice architecture for making people's choices more climate-friendly, this approach may unlock new pathways for higher liveability of cities.

Climate change poses an urgent challenge, necessitating drawing on strategies and policies on all levels to mitigate its impacts and advance global climate action. Within this context, the behavioral sciences—particularly behavioral economics and psychology—offer valuable insights into how human behavior can be guided towards more sustainable practices, pointing to structural actions by governments, midlevel decisions by business, and individual decision-making[1]. These disciplines emphasize the role of choice architecture: structures or systems designed to influence individuals' decisions predictably without directly targeting people's preferences nor intruding into their freedom of choice.

A critical yet often overlooked aspect of choice architecture is urban form—the physical layout and design of cities—which significantly influences mobility behaviors, energy consumption in buildings, and even food choices[2]. Traditionally, the domain of urban and transport planning, grounded in engineering and planning disciplines, has not extensively incorporated behavioral insights. This gap represents a missed opportunity for synergizing knowledge to create choice environments that naturally encourage low-carbon behaviors across consumption domains such as mobility, living, and food.

## Effective behavioral interventions for climate action

Behavioral economics integrates insights from psychology into economic models to understand decision-making processes. It reveals that individuals are "boundedly rational"[3], do not always act in their long-term self-interest, and are highly context-dependent in their decisions. This perspective is particularly potent for addressing climate change and sustainability challenges, where individual choices collectively exert significant impacts on the environment. Behavioral interventions, subtly modifying the decision-making environment (architectural nudges) or providing relevant information in accessible ways (educational nudges), can steer individuals towards more sustainable choices without restricting freedom of choice[4] (for examples, see Box 1).

Interventions like the above, which leverage inherent biases and heuristics in human decision-making, have shown effectiveness and acceptability in various contexts and across nations. They demonstrate the potential of behaviorally informed policies to overcome barriers to change,

create habits through affordances, and facilitate small but significant shifts in individual behavior across large populations.

## Planning and engineering profits from behavioral insights

Urban and transport planning traditionally focuses on the physical and infrastructural elements of cities and transport systems, aiming to optimize urban form and mobility for efficiency, accessibility, and sustainability. Despite some implicit understanding of human behavior, planners often do not systematically integrate insights from behavioral disciplines into their strategies. This can lead to a mismatch between how environments are designed and how they are used in practice.

Urban and transport planning focuses on the "6Ds": population Density, Diversity of land use, Design of the street network, Distance to transit, Destination accessibility, and Demand management. Planners use these six variables to shape urban form and transport systems, aiming to reduce reliance on private vehicles and encourage low-carbon mobility options such as walking, cycling, and public transport. The concept of the 15-minute city, where essential services and amenities are within a 15-minute walk or cycle from home, is a prime example of designing urban spaces that theoretically promote sustainability by reducing the need for longer-distance travel.

These approaches often originate from a planning and engineering mindset. They focus on altering the built environment and providing alternative transport options but may not fully account for the complex web of factors influencing individual needs and behaviors. Behavioral insights—such as the impact of choice contexts, social norms, cognitive biases, and other perceived barriers to behavior change—are rarely systematically integrated into the planning process. However, a well-curated choice environment makes the most sustainable choice the most affordable, available, accessible, and attractive choice (the "Four As"). Without a behaviorally informed view, even well-designed urban spaces and transport systems may not achieve their full potential for promoting low-carbon mobility if they do not resonate with the psychological and social realities of the people who use them.

## Lighthouse cities use a design mindset for people

Behavioral Design Thinking in urban design bridges the gap between behavioral insights and technical planning, emphasizing empathy, iterative prototyping, and a user-centered approach.

This approach has gained prominence through the influential work of Jan Gehl, a renowned Danish architect and urban design consultant. Gehl has significantly influenced the field of urban design with his focus on making cities more livable and sustainable by prioritizing human needs. Jan Gehl's principles focus on designing cities for people, suggesting that understanding how people use and interact with spaces is crucial for creating vibrant, liveable, and sustainable urban environments[5]. Several cities

## Box 1 | Examples of behavioral interventions for sustainability

**Default Choices**: Setting eco-friendly options as defaults in various settings (e.g., green energy for utilities) significantly increases the uptake of these options. This effectiveness largely stems from the "status quo bias," where individuals are likely to stick with pre-set options due to inertia, perceived endorsement, or the perceived effort in changing them. An illustrative case is renewable energy defaults in Germany's and Switzerland's energy transformation. When green energy tariffs became the default choice, energy providers witnessed a substantial increase in uptake, compared to countries that required active choice in opt-in schemes[10].

**Social Norms Feedback**: Our friends' and neighbors' choices profoundly affect our own. Providing feedback on energy consumption by comparing an individual's usage with their significant peers can motivate individuals to reduce their energy use. This leverages the powerful influence of social norms and status-seeking. Most people seek social approval from their respective significant social reference group by adhering to the socially expected behavior. A notable example is the

Opower program, implemented in various regions, including the United States. It has consistently shown reduced energy consumption among households receiving social norm feedback on their utility bills. Another example is the well-documented "behavioral contagion" of (highly visible) solar modules on homes in communities.

**Salience and Framing**: Making the environmental impact of choices more visible and framing options in a way that highlights their environmental consequences can influence decision-making. For instance, including carbon footprint information on product labels or menus can nudge consumers towards lower-carbon choices. This approach capitalizes on the tendency for individuals to give greater weight to more salient information. In Sweden, including carbon footprint labels on grocery items has increased the purchase of products with lower environmental impacts. Apps and gamification, if well designed, can make sustainability issues salient (e.g., the 'Mordor Sharper' game and the 'Kids-go-green' app), which in turn leads to measurably less energy-intensive behavior[11].

## Box 2 | How cities make use of design thinking

Copenhagen's transition towards prioritizing bicycles over cars didn't just involve constructing bike lanes; it was underpinned by a deep understanding of behavioral incentives. For instance, Copenhagen's bike lanes are designed with attention to details that matter to cyclists—such as direct routes that make cycling the fastest mode of transport for short trips, physical separation from car traffic for safety, and even green waves for bikes during peak hours to minimize stopping at red lights. These features directly appeal to the behavioral tendency for choosing the path of least resistance, making cycling an easy, safe, convenient and, most importantly, a time-saving choice. The new Carlsberg quarter took Gehl's lessons up and designed around the movement of people, not cars (Fig. 1).

In turn, Melbourne's revitalization of its laneways was based on the insight that people are drawn to vibrant, human-scale spaces. By

transforming these spaces into areas filled with art, cafes, and greenery, the city leveraged the human inclination towards socializing and exploring intriguing environments. This intervention was grounded in the understanding that environments can shape social interactions and encourage more pedestrian activity, thereby reducing the reliance on cars.

Bogotá's Ciclovía program's success is a testament to designing urban initiatives that tap into latent desires for community engagement and physical activity. By closing streets to motor traffic on Sundays, the city not only provides safe spaces for exercise but also fosters a sense of community and belonging. This initiative understands and leverages the social aspect of behavior, encouraging people to participate in physical and social activities outside of their homes, which also reduces vehicle emissions.

worldwide have implemented this approach, making them attractive exemplary lighthouses for human-centred urban design. Meticulous observation and analysis of human behavior informed the construction of pedestrian-friendly spaces, human-scale buildings, and accessible public areas that encourage social interaction and active modes of transportation.

Some of the world's most sustainable lighthouse cities, such as Copenhagen, Melbourne, and Bogotá made use of design thinking and behavioral insights (Box 2). In each of these cities, the application of behavioral insights goes beyond the physical infrastructure to the subtle cues and amenities that encourage desired behaviors. Unlike traditional urban planning, which might focus primarily on the functional aspects of infrastructure, these interventions are designed with a keen understanding of human psychology, aiming to nudge people towards more sustainable and health-promoting behaviors by making those behaviors more accessible, attractive, and enjoyable. During planning stages, people fear change and often perceive it as threatening the status quo. However,

as the COVID-induced closing of streets and redesign of neighbourhoods in many cities worldwide show, people quickly adapt to structural changes in their surroundings and, more often than not, appreciate the gains they deliver once implemented. In this sense, COVID constituted a natural living lab.

### Overcoming barriers

In urban planning, utilising behavioral insights encounters significant barriers, prominently due to the low behavioral literacy among municipal and council-level actors. One of the more daunting hurdles is the perceived and real high cost associated with implementing infrastructure changes. This is compounded by the uncertainty surrounding the cost-effectiveness of behavioral nudges (e.g., for the case of encouraging physical activity[6]). Moreover, many municipalities lack a culture that supports experimental approaches to policy implementation and are limited by governance structures that lead to compartmentalized thinking. Tellingly, a dedicated

## Box 3 | Five application areas for urban design thinking for sustainability

1. **Building Codes**: Modern building codes should mandate designs that prioritize sustainable travel modes right from the entrance areas. For example, easy access to bike storage, visible and attractive walking paths, and charging stations for electric vehicles can encourage the use of these modes over traditional car use. Behavioral insights can inform how visibility, convenience, and perceived safety can influence the choice of travel mode, making sustainable options the default choice for building users.

2. **Street and Transport Network Design**: Streets should be conceptualized not just as conduits for vehicles but as multifunctional spaces that encourage walking, cycling, and social interaction. This requires understanding the psychological effects of street width, pavement quality, the presence of greenery, and the arrangement of street furniture. Behavioral insights into how these factors influence feelings of safety, comfort, and inclination to choose active modes of transport can lead to more effective street designs. An example is the understanding of behavioral factors in the uptake of cycling in the context of urban design.

3. **Public Spaces**: The design of public spaces can significantly influence how they are used and the types of activities they promote. Incorporating features that encourage physical activity, social interaction, and connection with nature can enhance well-being and environmental sustainability. Behavioral science can provide insights into the types of designs and features that draw people to these spaces and encourage prolonged use, such as the placement of seating, water features, and the programming of regular community events.

4. **Apps and Digital Devices**: The way individuals interact with apps and digital devices to make travel decisions profoundly impacts urban mobility patterns. Designing and improving integrated multimodal apps and data platforms, such as Berlin's Jelbi, with behavioral insights in mind can nudge users towards more sustainable modes of transport. Features such as highlighting the environmental benefits of certain choices, making sustainable options more prominent by putting them first on the results list, or providing social rewards for low-carbon travel can leverage cognitive biases to promote greener travel decisions. People's decisions are heavily influenced by earlier experiences; personalized search results help overcome initial barriers to change.

5. **Communication Strategies**: Effective communication between municipal agencies and citizens is crucial for promoting sustainable urban transport modes. Behavioral insights can inform the development of communication strategies that prime new residents with information on low-carbon modes of transport, utilize social proof to highlight the popularity of sustainable travel options, and frame messages in ways that resonate with individuals' values and identities. For example, tailored welcome packages for new residents that include maps of walking and cycling routes, public transport schedules, and incentives for trying out these modes can set the stage for sustainable travel patterns. For some targets, belongingness to a neighbourhood/city and appeals to shared memories of a "greener past" can be effective.

behavioral unit within the mayor's office often proves essential for building the necessary competencies to apply these insights effectively[7]. The municipality of Copenhagen, for instance, has skilled up their staff by attending external courses and cooperating in concrete urban planning projects with external behavioural experts and consultancies. In contrast, the lack of a robust infrastructure for knowledge transfer and the absence of strong advocacy can hinder the uptake of successful models from more pioneering cities. This issue is exacerbated in municipalities overwhelmed by urgent socioeconomic challenges, where behavioral strategies may be perceived as less critical compared to immediate needs like infrastructure or social services.

A key challenge is the diffusion of successful innovative practices from lighthouse cities to cities with fewer resources. One approach is to foster "behavioral literacy" in the municipalities, by integrating behavioral insights skills into the curricula and professional development programs for urban planners. Competently applying a "behavioural lens" in various planning phases enables planners to analyse urban mobility from a systems and human-centred perspective, which goes far beyond a limited "bike versus car" debate. Enhancing the diffusion of behavioral strategies in cities with limited capacity could be supported by high-visibility international events or competitions of cities, boosting their attractiveness and branding, providing a platform for showcasing and learning from each others' innovative practices.

Similarly, the high visibility of the Intergovernmental Panel on Climate Change (IPCC) in practical policymaking makes it an important arena for providing evidence and insights. The previous IPCC's report on climate solutions underscored the significance of behavioral solutions and the role of urban infrastructures in mitigating climate change[8]. Despite highlighting these crucial areas independently, the report stopped short of weaving them into a cohesive assessment that combines the influence of urban form on human behavior with the principles of behavioral sciences. We suggest this oversight presents a clear gap in the current literature and interdisciplinary practice. However, the upcoming IPCC special report on cities emerges as a promising opportunity to bridge this divide. By systematically integrating the insights from behavioral sciences with the strategic development of urban infrastructures, this forthcoming report could set a new precedent for interdisciplinary approaches to climate action, working out synergies between urban planning and behavioral interventions.

### Scaling sustainable urban design by integrating behavioral disciplines

Scaling behavioral insights into urban planning will require mainstreaming the underlying insights into education. While manuals and frameworks for incorporating behavioral sciences into urban policy-making exist, such as the Behavioural Insights Team's various guides or the World Bank's "Mind, Society, and Behavior" report, a focused manual for urban planning remains underdeveloped. A behaviorally updated manual would serve as a bridge between the theoretical understanding of human behavior and the practical applications in urban environments. It would provide examples and guidelines to help planners and designers create attractive and sustainable spaces (see Box 3 for examples) and facilitate collaboration by establishing a common language and understanding.

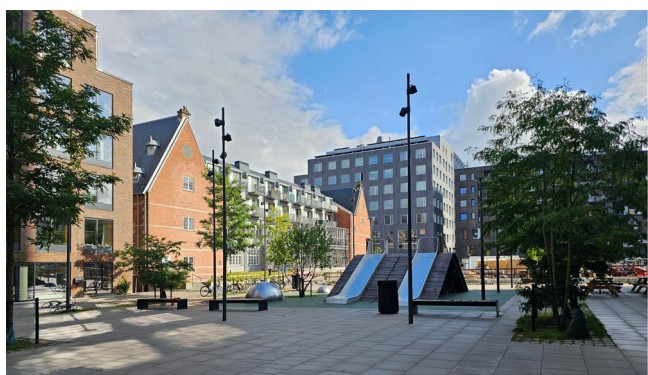

**Fig. 1 |** Carlsberg Byen, Copenhagen. The new quarter is designed for movement at human scale. Photo: Felix Creutzig.

Emerging big data will be helpful in translating behavioral insights from manuals into practice. For example, user behavior in co-working spaces can be observed to design co-working spaces better. Similarly, observed travel behaviors and their contexts can serve as a basis for designing streets that nudge users towards sustainable modes of transport. Integrating existing big data approaches and detailed data-based understanding of how citizens move in urban environment, tracking movements, not people and their identities, hence holds promise for urban planning.

## Towards holistic urban planning

Integrating behavioral sciences into urban planning represents an important shift towards scaling sustainable urban environments, enhancing the liveability, sustainability, and resilience of cities by designing urban spaces and systems that are in harmony with human behavior. Realizing this potential, however, requires overcoming institutional barriers and disciplinary silos that have traditionally separated planners, architects, engineers, and behavioral scientists. It also requires supplementing a market-based preference approach with an idea of additionally providing infrastructures aimed at improving objective measures of well-being[9].

The participative co-design and dissemination of a comprehensive manual that outlines practical strategies for incorporating behavioral insights into urban planning can serve as a critical tool in this endeavor. Such a document outlining how to "put people centre stage" would have a threefold benefit: it could help avoid the typical behavioral pitfalls of planning for change, such as the endowment effect and status quo biases; it would facilitate the sharing of knowledge across disciplines and between practitioners and researchers; and it would thirdly provide a clear framework for action, encouraging the adoption of a more human-centric approach in urban design and policy-making. Providing such guidance would be a valuable task for the upcoming IPCC's special report on cities.

**Felix Creutzig** [ID][1,2,3] ✉ **& Lucia A. Reisch** [ID][4]

[1]Technical University Berlin, Berlin, Germany. [2]Mercator Research Institute on Global Commons and Climate Change, Berlin, Germany. [3]Bennett Institute for Innovation and Policy Acceleration, University of Sussex Business School, Brighton, UK. [4]El-Erian Institute, Cambridge Judge Business School, University of Cambridge, Cambridge, UK. ✉e-mail: creutzig@tu-berlin.de

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

## Acknowledgements

The work has been supported by the YNOT Institute at Queens' College, Cambridge. FC acknowledges funding from the Horizon Europe Research and Innovation Action Programme under Grant Agreement No. 10105681. The funders had no role in the preparation of the manuscript or decision to publish.

## Author contributions

FC and LR conceptualized the paper. FC wrote the first draft. FC and LR edited the document in several rounds.

## Funding

## Competing interests

The authors declare no competing interests.
