## [transparent peer review · Communications Psychology]

Decision letter and referee reports: first round

Dear Professor Creutzig,

Your Comment titled "Bridging Behavioral Sciences and Urban Planning: A Joint Research Agenda for Climate Action" has now been seen by 2 referees, whose comments appear below. In light of their advice, I am delighted to say that we are happy, in principle, to publish it in Communications Psychology.

We will not send your revised paper for further review if, in the editors' judgment, the referees' comments on the present version have been addressed. If the revised paper is in Communications Psychology format, in an accessible style, and of appropriate length, we shall accept it for publication immediately. I have attached an edited version of your manuscript, and ask you to attend to each comment in detail.

EDITORIAL REQUESTS:

* Please check whether your manuscript contains third-party images, such as figures from the literature, stock photos, clip art or commercial satellite and map data. If any of the display items in your manuscript (figures, tables, boxes or movies) include images that are the same as, or are adaptations of, previously published images, please fill in the Third Party Rights Table, and return to us when you submit your revised manuscript. This information will enable us to obtain the necessary rights to re-use such material. If we are unable to obtain the necessary rights to use or adapt any of the material that you wish to use, we will contact you to discuss alternative options.

* Communications Psychology uses a transparent peer review system. On author request, confidential information and data can be removed from the published reviewer reports and rebuttal letters prior to publication. If you are concerned about the release of confidential data, please let us know specifically what information you would like to have removed. Please note that we cannot incorporate redactions for any other reasons.

*If you have not done so already, please alert me to any related manuscripts from your group that are under consideration or in press at other journals, or are being written up for submission to other journals (see

www.nature.com/authors/editorial_policies/duplicate.html for details).

FORMATTING GUIDELINES:

** Preface

The paper's preface (up to 40 words; without references) should serve both as a general introduction to the topic, and highlight your position or proposal. Because we hope that researchers across all fields of psychology will be interested in your work, the preface should be as accessible as possible.

** Length

The ideal length for Comment article in Communications Psychology is 1,500 words. We have some flexibility, however, but please ensure that your text does not exceed 1,800 words.

** Main text

* References

References appear as superscript Arabic numerals, in order of mention. The reference list mentions references in the numerical order in which they are mentioned in the main text. If a reference is cited more than once, the same number is used throughout the text and the reference receives a single entry in the reference list.

Only papers that have been published or accepted by a named publication should be in the reference list (preprints and citations of datasets are also permitted).

Unpublished/Submitted research should not be included in the reference list; it should only be mentioned briefly and parenthetically in the main text. Note that no major arguments should rely on unpublished research.

Published conference abstracts and URLs for websites should be cited parenthetically in the text, not in the reference list.

Footnotes are not used.

* Competing interests

Please include a "Competing interests" statement after the References. Note that we ask authors to declare both financial and non-financial competing interests. For more details, see <https://www.nature.com/authors/policies/competing.html>. If you have no financial or non-financial competing interests, please state so: "The authors declare no competing interests."

SUBMISSION INFORMATION:

In order to accept your paper, we require the following:

- * A cover letter describing your response to our editorial requests.
- * A separate document detailing your point-by-point response to any issues raised by our referees (please include the referees' comments in this document).
- * The final version of your text as a Word or TeX/LaTeX file, with any tables prepared using the Table menu in Word or the table environment in TeX/LaTeX and using the 'track changes' feature in Word.
- * Production-quality versions of all figures, supplied as separate files. Photographic images should be 300 dpi in RGB format (.jpg, TIFF or native Photoshop format) and any labels/scale bars included in a separate layer from the image. Line art, graphs and schemes should be vector format (.ai, .eps, .pdf); Adobe Illustrator files are preferred and will minimize production time. Any chemical structures or schemes contained within figures should additionally be supplied as separate Chemdraw (.cdx) files.

Communications Psychology is a fully open access journal. Articles are made freely accessible on publication. For further information about article processing charges, open access funding, and advice and support from Nature Research, please visit <https://www.nature.com/commpsychol/open-access>

Please note that your paper cannot be sent for typesetting to our production team until we have received this information; **therefore, please ensure that you have this ready when submitting the final version of your manuscript.**

ORCID

Communications Psychology is committed to improving transparency in authorship. As part of our efforts in this direction, we are now requesting that all authors identified as ‘corresponding author’ create and link their Open Researcher and Contributor Identifier (ORCID) with their account on the Manuscript Tracking System (MTS) prior to acceptance. ORCID helps the scientific community achieve unambiguous attribution of all scholarly contributions. For more information please visit <http://www.springernature.com/orcid>

For all corresponding authors listed on the manuscript, please follow the instructions in the link below to link your ORCID to your account on our MTS before submitting the final version of the manuscript. If you do not yet have an ORCID you will be able to create one in minutes.

IMPORTANT: All authors identified as ‘corresponding author’ on the manuscript must follow these instructions. Non-corresponding authors do not have to link their ORCIDs but are encouraged to do so. Please note that it will not be possible to add/modify ORCIDs at proof. Thus, if they wish to have their ORCID added to the paper they must also follow the above procedure prior to acceptance.

To support ORCID's aims, we only allow a single ORCID identifier to be attached to one account. If you have any issues attaching an ORCID identifier to your MTS account, please contact the Platform Support Helpdesk.

Please use the following link to submit the above items:
(link Redacted)

We hope to hear from you within two weeks; please let us know if the process may take longer.

Best regards,

Marike Schiffer

Marike Schiffer, PhD
Chief Editor
Communications Psychology

REVIEWERS' COMMENTS:

Reviewer #1 (Remarks to the Author):

First of all, the paper presented is timely and of general interest to the reader/community. It is well written and the story is well defined, presented and clearly structured. The points which are risen by the authors are relevant to the field and relevant literature is cited. The topic itself is neither specifically new nor it presents an innovative design or methodology. That however is not really necessary, because it brings together rather simple ideas with the rather complex field of urban planning. That story is being put nicely and also substituted by good examples.

That however is only one side of the coin, as which is most common to these kind of studies it misses the "last mile" of solution, as it stops at the more or less theoretical aspects. That means, how these insights can be actually put into action is not answered. More insights on how these ideas can be properly translated into action would help to complete the story. Saying that, the paper offers a good starting point/ raises valuable points addressing sustainable urban planning and is designed as compact as it can be for getting the intention, however I would like to address some points which might be possible to be clarified in a revised version.

It is not entirely clear, if the core goal should be to guide inhabitants towards sustainable practices or if inhabitants themselves help designing new pathways. The key question here is: who is the body to "lead" this guidance"? This refers back to the question: how does a municipal administrative body, the city structure has to look like, that these ideas will come into action. Three example cities are mentioned, but however only the Bogota case offers a concrete program which has been installed. In the end, it is the cities internal structure and the mandate for climate action which hinders or fosters sustainable development as explained here.

In the following I will raise specific points:

- Is default choice that easy in practice? How does the "free-market" system work with

that? How should create/define that default choice?

- How are the “social norms” defined which you are referring to. Can that be standardized at all?
 - I wonder if it is that easy to use covid as good example. I understand the intention, but maybe one or two examples of a successful “living lab” would be interesting
 - Salience and Framing: is it possible to name a more local/detailed example
 - Line 130-133: With regard to the three “lighthouse” examples: how did these behavioural units look like inside the mayor offices? Who could that strong advocacy be?
 - Lastly: It would be interesting to elaborate a bit on the indented or existing translation of the study points or the IPCC special report into practical action, also in order to provide a more substantial answer on the way of bridging the gap between science and practical planning ◊ maybe that can be highlighted even on the three examples.
- Anyhow, I find the paper interesting and would be happy for further discussion on that.

Reviewer #2 (Remarks to the Author):

The Comment “ Bridging Behavioral Sciences and Urban Planning: A Joint Research Agenda for Climate Action ” focuses on integrating behavioral sciences with urban planning to enhance climate action and sustainability. The authors argue that urban planning and design, traditionally dominated by engineering and infrastructure perspectives, can significantly benefit from incorporating insights from behavioral economics and psychology. These disciplines offer valuable strategies, such as choice architecture, to influence human behavior towards more sustainable practices without limiting individual freedom.

Does the Comment formulate a novel, thought-provoking take? Yes. The Comment formulates a novel and thought-provoking take by proposing a systematic integration of behavioral sciences with urban planning to create sustainable, resilient and livable cities; opening new pathways for climate action through the strategic design of urban environments. A central topic in urban climate change adaptation.

Is the argument persuasive? The Comments is clearly written and well organized. However, I think it is not persuasive enough. The style in which it is written seems very linear. It enumerates and inform, but it does not persuade, nor convince. It reads like a report.

Does the Comment reflect a viewpoint that has recently or historically not received

sufficient exposure? Yes. But I suggest the authors to go much more in depth with idea of holistic approach better explaining the integration, or perhaps even blending, of behavioral sciences and psychology into planning. Also, making a graph to show visually what is now explained with words can help.

Will the Comment be of interest to researchers in your field, or a wider audience of readers? I think that this comment is too general to be of interest of researchers in my field, but it could be of interest for a broader audience.

Reply to the Reviewer comments

We appreciate the reviewer remarks. Find below a point by point response.

Reviewer #1 (Remarks to the Author):

First of all, the paper presented is timely and of general interest to the reader/community. It is well written and the story is well defined, presented and clearly structured. The points which are risen by the authors are relevant to the field and relevant literature is cited. The topic itself is neither specifically new nor it presents an innovative design or methodology. That however is not really necessary, because it brings together rather simple ideas with the rather complex field of urban planning. That story is being put nicely and also substituted by good examples. That however is only one side of the coin, as which is most common to these kind of studies it misses the "last mile" of solution, as it stops at the more or less theoretical aspects. That means, how these insights can be actually put into action is not answered. More insights on how these ideas can be properly translated into action would help to complete the story. Saying that, the paper offers a good starting point/ raises valuable points addressing sustainable urban planning and is designed as compact as it can be for getting the intention, however I would like to address some points which might be possible to be clarified in a revised version.

Many thanks for thoughts and suggestions. Please see below for specific responses.

It is not entirely clear, if the core goal should be to guide inhabitants towards sustainable practices or if inhabitants themselves help designing new pathways. The key question here is: who is the body to "lead" this guidance"? This refers back to the question: how does a municipal administrative body, the city structure has to look like, that these ideas will come into action. Three example cities are mentioned, but however only the Bogota case offers a concrete program which has been installed. In the end, it is the cities internal structure and the mandate for climate action which hinders or fosters sustainable development as explained here.

We had to be brief. The other cities mentioned also have specific programs. We added a sentence on Copenhagen (on the new Carlsberg quarter) for specificity.

In the following I will raise specific points:

- Is default choice that easy in practice? How does the "free-market" system work with that? How should create/define that default choice?

This is a deeper point. We started discussing this important question in Mattauch et al. 2016, and there is more work in preparation in our labs. The short answer is that the "free market" does not consider infrastructures, on principle, and that – given that preferences are endogenous – a key urban design task is to provide infrastructure that addresses objective measures of well-being. It will always be a tense relationship between free-market liberalism and soft paternalism which must be negotiated and decided on by the respective municipalities, ideally in a participative process with citizens/users.

Mattauch, L., Ridgway, M., & Creutzig, F. (2016). Happy or liberal? Making sense of behavior in transport policy design. *Transportation Research Part D: Transport and Environment*, 45, 64-83.

We added a sentence in the conclusion section reading: “It also requires supplementing a market-based preference approach with an idea of additionally providing infrastructures aimed at improving objective measures of well-being¹⁰.”

- How are the “social norms” defined which you are referring to. Can that be standardized at all?

Excellent point. Social norms will always depend on the respective relevant reference group. What is common to most people (hence, the approach per se can be standardized, but the norm itself and the norm leaders cannot) is that they want to belong. People are “social animals” and fear being socially excluded. Hence, people tend to adhere to social expectations and copy significant others’ behaviour and choices.

We now *extended* the example as follows:

Social Norms Feedback: *Our friends’ and neighbors’ choices profoundly affect our own. Providing feedback on energy consumption by comparing an individual’s usage with their significant peers can motivate individuals to reduce their energy use. This leverages the powerful influence of social norms and status-seeking. Most people seek social approval from their respective social reference groups by adhering to socially expected behaviour. A notable example is the Opower program, implemented in various regions, including the United States. It has consistently shown reduced energy consumption among households receiving social norm feedback on their utility bills. Another example is the well-documented “behavioral contagion” of (highly visible) solar modules on homes in communities.*

- I wonder if it is that easy to use covid as good example. I understand the intention, but maybe one or two examples of a successful “living lab” would be interesting

We cannot think of a better example than the COVID-induced temporary and short-notice closing and redesign of neighborhoods. COVID showed the adaptability of behavior and had a disruptive effect on urban design and transport planning, which are still in transformation and negotiated in communities to date. To the best of our knowledge, most cities did not just return to their initial design; most kept elements of COVID-induced structural changes that citizens learned to appreciate after having used the new structures for many months. In this sense, COVID constituted a natural living lab.

We added a sentence:

However, as the COVID-induced closing of streets and redesign of neighbourhoods *in many cities worldwide* show, people quickly adapt to structural changes in their surroundings and, more often than not, appreciate the gains they deliver once implemented. In this sense, COVID constituted a natural living lab.

PS we would have loved to expand further on living labs, but need to keep the text concise.

- Salience and Framing: is it possible to name a more local/detailed example

We added a sentence with a specific example here, reading: “Apps, games and gamification, if well designed, can make sustainability issues salient (e.g., the ‘Mordor Sharper’ game and the ‘Kids-go-green’ app), which in turn leads to measurably less energy intensive behaviour⁶.”

- Line 130-133: With regard to the three “lighthouse” examples: how did these behavioural units look like inside the mayor offices? Who could that strong advocacy be?

This is a very relevant question, indeed. However, we must be brief and cannot go into details. Luckily, some work on these units has been published, also by our own labs. We chose one of the three lighthouse cities and added one sentence:

The municipality of Copenhagen, for instance, has skilled up their staff by attending external courses and cooperating in concrete urban planning projects with external behavioural experts and consultancies

- Lastly: It would be interesting to elaborate a bit on the indented or existing translation of the study points or the IPCC special report into practical action, also in order to provide a more substantial answer on the way of bridging the gap between science and practical planning ◊ maybe that can be highlighted even on the three examples.

As this is a comment, we can't fully assess the feasibility and potential of practical options (and we give already a number of examples). As the last paragraph, we conclude with what academics should do next. We now add a sentence, suggesting that this could be a crucial task for the SR cities, reading: “Providing such guidance would be a valuable task for the upcoming IPCC’s special report on cities.”.

Anyhow, I find the paper interesting and would be happy for further discussion on that.

Reviewer #2 (Remarks to the Author):

The Comment “Bridging Behavioral Sciences and Urban Planning: A Joint Research Agenda for Climate Action” focuses on integrating behavioral sciences with urban planning to enhance climate action and sustainability. The authors argue that urban planning and design, traditionally dominated by engineering and infrastructure perspectives, can significantly benefit from incorporating insights from behavioral economics and psychology. These disciplines offer valuable strategies, such as choice architecture, to influence human behavior towards more sustainable practices without limiting individual freedom.

Does the Comment formulate a novel, thought-provoking take? Yes. The Comment formulates a novel and thought-provoking take by proposing a systematic integration of behavioral sciences with urban planning to create sustainable, resilient and livable cities; opening new pathways for climate action through the strategic design of urban environments. A central topic in urban climate change adaptation.

Is the argument persuasive? The Comments is clearly written and well organized. However, I think it is not persuasive enough. The style in which it is written seems very linear. It enumerates and inform, but it does not persuade, nor convince. It reads like a report.

Does the Comment reflect a viewpoint that has recently or historically not received sufficient exposure? Yes. But I suggest the authors to go much more in depth with idea of holistic approach

better explaining the integration, or perhaps even blending, of behavioral sciences and psychology into planning. Also, making a graph to show visually what is now explained with words can help.

Will the Comment be of interest to researchers in your field, or a wider audience of readers? I think that this comment is too general to be of interest of researchers in my field, but it could be of interest for a broader audience.

We acknowledge that we have not written to the satisfaction of this reviewer. However, from the comments it is hard to discern how we could actually improve the manuscript. We are already at the edge in terms of lengths and details. Providing more depth would mean to move from comment to full article, which we don't aim to do.